# Qualitative Insights into Key Angelman Syndrome Motor Related Concepts Reported by Caregivers—A Thematic Analysis of Semi-Structured Interviews

**DOI:** 10.3390/children10091462

**Published:** 2023-08-28

**Authors:** Miranda Rogers, Stéphane Motola, Yacine Bechichi, Céline Cluzeau, Tanguy Terray, Allyson Berent, Jennifer Panagoulias, Jessica Duis, Damien Eggenspieler, Laurent Servais

**Affiliations:** 1Department of Paediatrics, MDUK Oxford Neuromuscular Centre & NIHR Oxford Biomedical Research Centre, University of Oxford, Oxford OX1 2JD, UK; miranda.rogers@paediatrics.ox.ac.uk; 2Sysnav Co., 27200 Vernon, France; stephane.motola@sysnav.fr (S.M.); yacine.bechichi@sysnav.fr (Y.B.); celine.cluzeau@sysnav.fr (C.C.); tanguy.terray@sysnav.fr (T.T.); damien.eggenspieler@sysnav.fr (D.E.); 3Foundation for Angelman Syndrome Therapeutics (FAST), P.O. Box 40307, Austin, TX 78704, USA; allyson.berent@gmail.com (A.B.); jennifer.panagoulias@cureangelman.org (J.P.); 4Section of Genetics and Inherited Metabolic Disease, Department of Pediatrics, Children’s Hospital Colorado, University of Colorado Anschutz Campus, Aurora, CO 80045, USA; jessica.duis@childrenscolorado.org; 5Department of Paediatrics, Neuromuscular Reference Center, University Hospital Liège, University of Liège, 4000 Liège, Belgium

**Keywords:** Angelman syndrome, qualitative research, patient-centered

## Abstract

Previous patient-centered concept models of Angelman syndrome (AS) are integral in developing our understanding of the symptoms and impact of this condition with a holistic perspective and have highlighted the importance of motor function. We aimed to develop the motor and movement aspects of the concept models, to support research regarding motor-related digital outcomes aligned with patients’ and caregivers’ perspectives. We conducted a qualitative analysis of semi-structured interviews of 24 caregivers to explore AS motor-related features, factors influencing them and their impact on patients and caregivers.The most impacted motor features were gait, walking and stair-climbing. Half of caregivers ranked motor symptoms as one of the most burdensome symptoms of AS. Caregivers frequently reported physical therapy, motivation, medical management and age as factors influencing motor function in AS and reported that impaired motor function affected both patients and caregivers. Measures of lower-limb motor function were identified as relevant to monitor drug effectiveness in AS. Caregivers discussed expected benefits of a digital outcome and potential issues with wearable technology in the context of AS. We propose a new motor function patient-centered concept model, providing insights for the development of relevant, motor-related, digital outcomes in AS.

## 1. Introduction

Angelman syndrome (AS) is a rare genetic neurodevelopmental disorder resulting from deficient expression of the maternally expressed UBE3A gene [1]. It is characterized by a wide variety of phenotypic features, including but not limited to developmental delay, seizures, severe speech deficits, sleep disturbances, gastrointestinal issues, behavioral and anxiety issues and a range of motor difficulties [2]. These motor impairments include ataxia, dyspraxia, and difficulties with planning and control. Difficulties in ambulation in particular have previously been reported, including delayed walking acquisition, severe ataxia and wide-based gait [2,3,4]. This leads to a specific gait pattern with severe unsteadiness, resulting in cautious progression and at times, multiple falls when patients navigate alone. Approximately 10% of patients will never gain ambulation [5]. 

Patient-centered concept models are vital to ensure that research outputs answer questions relevant to those affected by diseases. Producing these models enables the development of outcome measures that are relevant to patients, caregivers, drug developers and regulators, and are particularly important in AS given that multiple therapeutic options are currently in development [6]. Previous work using caregiver and clinical interviews to create patient-centered concept models for AS were conducted [7,8]. These concept models were developed using literature reviews and interviews with caregivers and clinicians and have been integral in developing our understanding of the symptoms and the impact of this complex disease. These have had a broad focus on all aspects of this multifaceted disease, rather than a focus on any particular domain. The first model divides the signs, symptoms and characteristics of AS into 5 categories, including ‘physical’. This category comprises several motor features, including mobility, other gross motor problems, tremors, muscular issues, fatigue, and injuries from falling [7]. The second model lists ten ‘AS defining concepts’, including ‘motor’. This concept includes delayed motor skills, regression in fine & gross motor skills with age, gait (wide based, ataxic with uplifted arms & pronated ankles), balance and tremulous movement of limbs. This model identifies that walking difficulties are one of the most challenging symptoms of AS, considering them as a target for future therapies. However, there is no identification of factors impacting motor function nor the effects of motor impairment on patients and caregivers [8].

The development of meaningful outcome measures in AS is important as a range of therapies are in various stages of development, including those targeting motor difficulties [6]. New objective endpoints could help better define drug effects in AS, as well as provide a reliable tool for clinicians to follow-up their patients. Further understanding is needed of what constitutes a meaningful change in ambulation for these patients, in order to develop better outcome measures to test emerging therapies.

Understanding the best way to assess changes in these ambulation abnormalities that is clinically meaningful to patients and caregivers is difficult, due to patients’ gross developmental delay, variability in clinical presentation of the disease, behavioral issues and the rarity of the disease itself. Motor function of AS patients is currently assessed by non-specific scales, such as Vineland Adaptive Behavior Scale-3 (VABS-3), the Modified Performance Oriented Mobility Assessment-Gait (mPOMA-G), the Bayley Scales of Infant and Toddler Development-4 (BSID-4) and the Functional Mobility Scale (FMS) [7,9,10,11,12,13]. As well as being non-AS specific, these scales present other limitations such as a ‘floor effect’ (that is raw scores well-below age-matched controls) seen in the BSID-4, and subjectivity, seen with the clinician- or caregiver-reported scales [7,9]. Furthermore, performance-based measures such as the BSID-4 and mPOMA-G are extremely difficult to administer to individuals with AS given the significant cognitive and behavioral impairment and dyspraxia that is generally present [7]. Recently, digital outcomes have shown great promise in neuromuscular diseases, with one continuous real-world measured outcome derived from a wearable technology (SV95C) approved for use as a secondary endpoint by the EMA in Duchenne’s Muscular Dystrophy [14,15]. A similar approach could be beneficial in Angelman syndrome or other conditions with severe neurodelopmental delay, as a wearable device collects passive motor activity and does not require active patient participation.

The objective of this study is to further develop the motor and movement aspects of the patient-centered AS concept model to support research regarding the development of new motor-related digital outcomes aligned with patients’ and caregivers’ perspectives. We aimed to use the two aforementioned concept models of AS, with semi-structured interviews with caregivers of AS, to inform the development of a concept model focused on understanding the motor features of AS. We also aimed to identify variables that could be measured by wearable technology and gather caregiver opinions on the meaningfulness of these variables.

## 2. Materials and Methods

A qualitative analysis, using a thematic approach, of semi-structured interviews of caregivers of individuals with AS was undertaken. Interviews were conducted on 9–10 December 2022 during the Foundation for Angelman Syndrome Therapeutics (FAST) Global Science Summit held in Miami, Florida.

### 2.1. Design and Participants

Participants included were caregivers above 18 years old, involved in the care of patients living with a confirmed diagnosis of AS, and attending the FAST Global Science Summit (where they were recruited). 

### 2.2. Ethical Considerations

Ethical review and approval were waived for this study by Pearl Institutional Review Board, Indianapolis, according to FDA 21 CFR 56.104 and 45CFR46.104(b)(2): (2) Tests, Surveys, Interviews on 15 November 2022. Caregivers were provided with information regarding the research aims, data use and sharing, to allow for informed decision making in participation of the study.

### 2.3. Study Materials

A semi-structured interview guide was developed through consultation with experts in AS including a clinician and an Angelman syndrome patient advocacy research organization (FAST-US). 

Sociodemographics of caregivers and individuals with AS were collected at the beginning of the interview. ‘Able to walk’ was defined as patient ability to ambulate on their feet, with or without assistance. Assistance was then assessed in walking autonomy, with options including ‘usually walks with human assistance’, ‘usually walks with aid of a device’ and ‘usually walks without human assistance or aid of a device’.

### 2.4. Recruitment and Interviews

24 interviews were performed, representing 24 different individuals with AS. All interviews were analyzed. No participants withdrew. Interviewees were composed of at least one caregiver. In some cases, there were 2 caregivers (in most cases both parents), and occasionally the individual with AS was also present (but they only gave episodic answers to very sporadic questions when prompted by caregivers). Demographic data on the caregiver in the case of 2 caregivers present was based on who was allocated as the ‘main’ caregiver and answered the majority of the questions. Based on previous elicitation work in AS, we expected that saturation of concepts would be reached with approximately 20 participants [8]. Interviews were conducted by two researchers, either in English, Spanish or French, lasted approximately 30 min, and were audio recorded. Only the interviewee(s), the individual with AS they care for, and the interviewer were present during the interviews. No repeat interviews were conducted, and field notes were not made. Recordings were transcribed verbatim. Transcripts were reviewed by the research team to remove identifying data and translated into English by a professional translator where appropriate and recording destroyed. Transcripts were not returned to participants.

The interviewers and those performing the analysis had previous experience in healthcare research, and were supported by researchers with experience in qualitative research. The participants were aware of the interviewers’ background and aims of research.

### 2.5. Analysis

Once all interviews had been completed, qualitative data was analyzed using thematic analysis in ATLAS.ti. Two researchers read all the transcripts, identified topics arising from the questionnaire and developed a framework for coding. The researchers coded the transcripts using the coding framework developed. The coding framework continued to be refined during the coding process using an iterative process, involving discussion between the reviewers during the coding and the addition of new codes. After the coding process was completed, all the transcripts were checked to ensure quality of coding. 

Data was then grouped into themes looking at the motor and movement aspects of the AS concept model. We considered the concepts as falling into one of three main themes—impaired motor features (symptoms or features of the disease, relating to motor movements), factors affecting motor function (such as age or behavior, and external factors, such as environment) and impact of impaired motor function (how the impaired motor features affect the lives of individuals with AS and caregivers).

Saturation analysis of concepts included in the concept model was conducted to ensure all relevant concepts were captured. We analyzed saturation of concepts arising in the transcripts by splitting the set of transcripts into quarters and comparing the concepts arising in each quarter. We only included concepts reported by greater than or equal to 10% of the interviewees (i.e., 3 or more) in the saturation and the overall analysis. The order of transcripts chosen for saturation analysis was according to availability of the transcripts.

## 3. Results

### 3.1. Sociodemographics

The sociodemographic characteristics of caregivers and individuals with AS are shown in Table 1. Of note, one clinician interview was included throughout (a consultant neurologist, who reported on an individual with AS she follows in her clinic).

### 3.2. Analysis of Concepts

#### 3.2.1. Saturation Analysis

We only included concepts reported by greater than or equal to 10% of the interviewees (i.e., 3 or more). 40 concepts arose in the first 6 interviews analyzed, with 2 further concepts arising in the next 6 (Table 2). No new concepts reported by ≥10% of the interviewees emerged after analysis of the first 12 interviews had been coded, confirming saturation was achieved and supporting the number of interviews used (Table 2). 

#### 3.2.2. Motor Features Impacted in AS

One of the topics raised was the burden of AS motor symptoms compared to other AS-related symptoms. Caregivers responded to ‘How burdensome is the impaired lower limb motor function compared to other AS symptoms?’ using a scale of 1–5, with 1 being the most burdensome symptom of AS and 5 being the least burdensome symptom of AS. Half of caregivers scored impaired lower limb motor function as a 1/5 or 2/5, implying they consider it to be the most or one of the most burdensome symptoms of AS (Figure 1). However, there was a range of results, highlighting the heterogenous nature of AS symptoms and the importance of considering other symptoms alongside motor function in AS presentation, assessment, and management.

When considering patient ambulation status, caregivers of non-ambulant patients ranked lower limb motor function between 1 and 3 (i.e., towards the most burdensome symptom of AS end of the spectrum), whilst caregivers of ambulant patients ranked it between 2 and 5 (i.e., towards the less burdensome symptom of AS end of the spectrum), highlighting a difference in caregiver perspectives (Appendix A).

The burden of motor symptoms was raised in response to this question and spontaneously throughout the interviews, with both descriptions of the high burden of motor function and descriptions of other AS-related features that were considered equally or more important. These included cognitive development, communication difficulties, fine motor and behavioral issues (Appendix A).

The 24 qualitative interviews identified fourteen motor features that caregivers considered affected by AS (Figure 2A). The most commonly reported affected motor feature was gait (23/24 interviewees), closely followed by walking, and climbing stairs (22/24 interviewees). Falls, running and balance were also commonly reported to be affected (21/24, 19/24 and 18/24 interviewees, respectively). In contrast, other features less commonly reported included impairment in jumping (4/25 interviewees) and using crawling instead of walking (4/24 interviewees). Example quotes for each motor feature are reported in Table 3. 

One interesting aspect that emerged was the interrelation between different features of AS. Multiple interviewees discussed the connection between motor function and other AS features, including communication and cognition. 

“Independence is driven by walking ability to be able to go up and interact with people. Even if he had a higher level of social and communication abilities, certainly, that would be limited if he was not able to approach people independently”.Clinician discussing an 8-year-old male, USA

“So the motor thing on its own means nothing. How can we separate just motor function?”Mother of a 5-year-old female, Chile

Furthermore, multiple caregivers reported the change in burden over time, with the motor function initially being a substantial burden, but this waning as the individual with AS gained some gross motor skills with age. 

“At the beginning, it was a very significant burden. Now that she’s managing to walk, not so much”.Mother of a 5-year-old female, Chile

#### 3.2.3. Factors Affecting Impaired Motor Function in AS 

Fourteen different factors were brought up by caregivers as affecting the motor function of individuals with AS (Figure 2B).

The most commonly reported factor was physical therapy (22/24 interviewees). Other forms of medical management beyond physical therapy (such as orthotics, medications, specialized equipment) was a further commonly reported factor (15/24 interviewees). Although many of these comments were positive or neutral, some caregivers reported negative impacts of medical management, in particular the negative effects of seizure medication (Table 3). Furthermore, several difficulties relating to medical management of AS emerged, including accessing medical care due to location, timeframe or specialists availability, lack of information on the condition and difficulty for patients to engage in care (Appendix A).

Motivation was also a commonly reported factor affecting motor function (18/24 interviewees), with caregivers often reporting that it meant the individual with AS would move faster, further or for a longer duration (Table 3).

A further commonly reported factor affecting motor function was age (14/24 interviewees). Many caregivers reported that increasing age was related to an improvement in motor function. However, some caregivers reported that as individuals with AS got older, there was a negative impact on motor function, with individuals reported as mobilizing slower and even regressing (Table 3).

A description of the remaining factors, along with illustrative quotes, is shown in Table 3.

#### 3.2.4. Impact of Impaired Motor Function 

The 24 qualitative interviews also investigated the impact of impaired motor function on different aspects of individuals with AS and caregivers’ lives (Figure 2C). The reported impacts affected both individuals with AS and caregivers. 

The most commonly reported impacts of impaired motor function were concerns for safety of individual with AS, the financial cost of AS, the effect on the independence of the individual with AS and the impact on the individual’s mood/emotions/behavior (18/24 interviewees). 

With regards to safety of individual with AS, caregivers reported concerns particularly around falls, navigating stairs, streets and new environments. It was raised that the development of motor skills such as walking, could increase rather than diminish safety concerns, due to increased independence and hence less caregiver control over individual’s actions. However, concerns for safety were reported by caregivers of both ambulant and non-ambulant individuals with AS, emphasizing that this is an issue for both groups of patients.

The financial cost of caring for an individual with AS was also commonly reported as an impact, and included costs for medical personnel, such as doctors and physical therapists, costs of home modification, costs of equipment and costs of care. This ties into the impact of job sacrifices made by caregivers in relation to the individual with AS (caregiver job trade-offs), including giving up formal employment or working fewer hours due to caring responsibilities. Interestingly, when discussing the financial impact of motor issues in AS, caregivers noted that the financial impact varies with the different features of the disease.

“For gastrointestinal issues, there are medications that are fully reimbursed… There is less of a financial impact on the communication aspect”Mother of a 1-year old female, France

A description of the remaining impacts on impaired motor function, with example quotes, is shown in Table 3.

#### 3.2.5. Relevance of Motor Function to Clinical Trials 

In response to the question: “In your perspective, which improvement of motor function would have the greatest impact on your family life?”, an improvement in walking was the most consistently raised concept (14/19 interviewees who responded). The responses included elaborations relating to walking independently, walking a normal distance, walking at a normal pace and walking with balance.

“I think he would be able to go at a normal pace for an hour, that would be pretty life-changing for us”.Mother of a 6-year-old male, USA

“If she could walk independently. Just being able to get up and walk independently would have the biggest impact”.Mother of a 22-year-old female, USA

However, some caregivers mentioned other motor concepts that were of particular importance to them, such as balance, falling, standing for long periods of time, fine motor skills, swimming and daily activities. 

“That [improvement of motor function] would not be the gross motor, but the fine motor skills. That’s the eating part”.Mother of a 4-year-old female, Belgium

Motor functions reported as impaired and discussed in response to this question have been highlighted in Figure 1 (starred bars). 

Caregivers were also asked to rate a range of potential lower-limb motor function variables as relevant or not relevant to monitor drug effectiveness in AS (Figure 3). Most caregivers rated most lower-limb motor function variables as relevant (Figure 3). Some caregivers went into more detail regarding their thoughts on why in particular falls, stair climbing, stride length, number of strides per day, distance walked, stride velocity and walking perimeter were relevant variables to measure (Appendix A). These quotes supported that these 7 measures reflect real-world functioning of individuals with AS. Caregivers also discussed other potential variables that they felt could be relevant, including variable relating to ataxia, direction, speed and fatigue (Appendix A). 

During this section of the interview, caregivers raised positives regarding the use of a wearable device, as well as potential issues. They described a lack of objective measures in current clinical outcomes for AS, as well as outlined the positives of having real-time data and dismissed concerns regarding wearability (Appendix A). However, they also described potential issues, including the effects of other factors affecting the measurement, including age, environment, tiredness, climate, medication and the other features of AS such as cognition (Appendix A). In particular, the difficulty in differentiating between assisted movement and unassisted movement was raised by multiple caregivers as a potential issue in using a real-world wearable technology, as many caregivers reported assistance (hand over hand) to have a major impact on AS motor functions. Furthermore, other caregivers commented on the tolerability of wearing the device (although this was considered not an issue by another interviewee), how to interpret the data of individuals with limited movement, and how to differentiate going up and down stairs as going down was generally reported as more difficult (Appendix A).

### 3.3. A Motor Function Concept Model of AS

Following on from the previous concept models, we developed a more in-depth model of motor function in AS from our caregiver interviews (Figure 4A). This model includes some motor concepts that emerged in the previous models, such as gait, mobility, falling and balance. Although our model goes into more detail regarding the factors affecting motor function and the impacts of motor impairment, there are a few aspects mentioned in the previous models that did not emerge in our interviews, such as tremors/tremulous movements.

Our work identified 6 commonly reported features of impaired motor function in AS (walking, climbing stairs, gait, falls, running and balance). Furthermore, variables assessing many of these, such as walking duration, stride velocity, falls and stride trajectory were reported by caregivers as relevant to monitor drug effectiveness in AS. Interviewees most consistently mentioned an improvement in walking as the motor concept that has the biggest impact on day-to-day life. These results, combined with one of the previous conceptual models of AS, have allowed us to develop a conceptual framework to help identify a concept of interest for the development of an outcome measure in AS (Figure 4B). 

## 4. Discussion

Our qualitative research has demonstrated the importance of motor features in the overall AS picture. We demonstrate a range of impaired motor features in AS, highlight that impaired motor function is clinically relevant, identify a wide range of factors affecting motor function, and discuss impacts of impaired motor function. These results emphasize that although AS is a multifaceted disease, the motor aspect is clinically meaningful and relevant to caregivers, and thus constitutes an important feature to monitor during therapeutic development.

Our concept model builds on the two previous concept models of AS, which had a much broader outlook at AS symptoms [7,8]. The sociodemographic characteristics of our caregiver cohort were similar to that of the previous models—these studies had a slightly larger number of interviews conducted (36 and 30 respectively), slightly older age of individuals with AS (mean 15.7 and 12.4, respectively) and similar spread of ethnicities [7,8]. However, one model only included ambulant patients, whilst the other included ambulant and non-ambulant but did not provide the precise respective proportions [7,8]. The motor concepts raised in the previous models generally overlap with those raised in our model: gait, balance, and walking were also raised in one or both of these studies [7,8]. Although our study went into more details and raised many other motor features not mentioned in these studies (as expected), some concepts raised in the previous models did not emerge in our analysis, such as tremors and oral motor impairment [7,8]. One possible explanation is that caregivers may not have considered tremors as a motor issue and hence did not mention it during our motor-focused discussion, highlighting a potential limitation of our study.

Although our interviews focused specifically on the motor functions of AS, the interrelation of other AS symptoms with motor features was demonstrated. When discussing the burden on motor function in comparison to other features of the disease, caregivers spontaneously highlighted the difficulty in separating out the effects of impaired motor function from other features of the disease, such as communication, fine motor skills and cognition. Furthermore, other features of the syndrome including seizures, cognition and communication impairment were elicited as factors affecting motor function. These findings highlight that motor function in AS cannot be considered in isolation, implying that motor function alone cannot be representative of AS as a whole, but also that motor function is important in the overall AS model. Furthermore, there is evidence from other neurodevelopmental disorders such as Noonan syndrome, Phelan-McDermid, spinocerebellar ataxia and Dup15q syndrome that the degree of motor impairment is linked to other disease domains, including cognition [16,17,18,19,20,21,22].

Our results identify that a range of different endpoints are clinically meaningful to patients. This supports the use of these different parameters as potential endpoints in AS. There is a need for the development of better outcome measures in AS, as many currently available measures are non-specific to AS, lack sensitivity and are inherently subjective [7,9]. Furthermore, understanding what is relevant to caregivers and patients is critical to ensure that new outcome measures focus on aspects important to the patient community [23]. The development of patient-focused concept models is central to achieve this goal, and in particular in this study, we received input from the patient advocacy group FAST-US to develop our interviews and protocol and to ensure that the content of our questions were meaningful to the individuals we aim to support.

Interviewees described a lack of objective measures in current clinical outcomes for AS, as well as outlined the benefits of having real-time data from a wearable device. Digital outcome measures attracted attention in the neuromuscular diseases field, with one outcome from a wearable device, SV95C, recently approved by the EMA as an outcome measure in clinical trials for Duchenne’s Muscular Dystrophy [14,15]. A recent study using pressure-sensor-based technology, inertial and activity monitoring, and instrumented gait analysis (IGA) in children with AS demonstrated that wearable technologies could identify differences in gait compared to neurotypical children, and could capture gait decline in children with AS over the course of the disease [24]. This suggests that wearable measures of gait could provide an objective outcome measure for this disease. Furthermore, this study suggests that wearable technologies may be acceptable to patients with AS, a concern raised by one interviewee. Furthermore, in AS natural history studies and clinical trials, a wearable gait sensor has been tolerated by 59/61 participants for at least the minimum hours of recording for analysis (50 h) [25]. Other concerns, such as differentiating between assisted and unassisted movements, and the impact of external factors on movement such as environment and medications, are important to consider in the development of such an outcome. Our model describing not only the impacts of impaired motor function but also the factors affecting it can help inform this development.

Strengths of this study include the thematic analysis, an established method of qualitative data analysis; evidence of concept saturation demonstrating that a suitable number of interviews were used to elicit appropriate concepts; and the input of the patient advocacy group FAST-AS in the development of our interviews and protocol, to ensure that the content of our questions were meaningful to the individuals we aim to support. There were several limitations to this study. Firstly, our caregiver cohort was selected from those attending the FAST Annual Global Science Summit, meaning that they may not be representative of the overall AS caregiver population. In addition, our caregiver cohort was relatively small and nearly exclusively composed of direct relations of individuals with AS, with one professional (consultant neurologist). Additional work encompassing more professionals involved in the care of AS would be valuable to further develop the concept model. The use of qualitative methods also introduces a degree of subjectivity to the analysis of the variables described by interviewees, so we used two researchers to conduct the coding. Future work should focus on the motor concepts raised in this study that have not been identified in previous work, to further quantify these with greater reliability and precision. Furthermore, as we chose to focus on AS motor features rather than start with a broader exploration of the patient and caregiver experience, there is the possibility that we may have missed some aspects of the context of the motor features within the wider AS symptoms. As mentioned earlier, some motor concepts reported in previous studies did not arise in our interviews, which may have been due to the purely motor-focused approach of our study. To mitigate this issue, we utilized the previously-reported conceptual models to develop our initial analysis plan, and highlighted the similarities and differences between their work and our findings.

## 5. Conclusions

We have explored in detail the motor features of AS from their caregiver perspective, developing an understanding of the impaired motor features and the factors affecting these and the impact of this impairment. We have developed a motor-focused concept model of AS, highlighting the importance of motor function in this disease. Furthermore, our work identified 6 important motor concepts (gait, walking, climbing stairs, falls, running and balance) which could serve as potential candidates for novel AS clinical endpoints and our work has provided context to support further investigation and development of these endpoints, using objective measures.

## Figures and Tables

**Figure 1 children-10-01462-f001:**
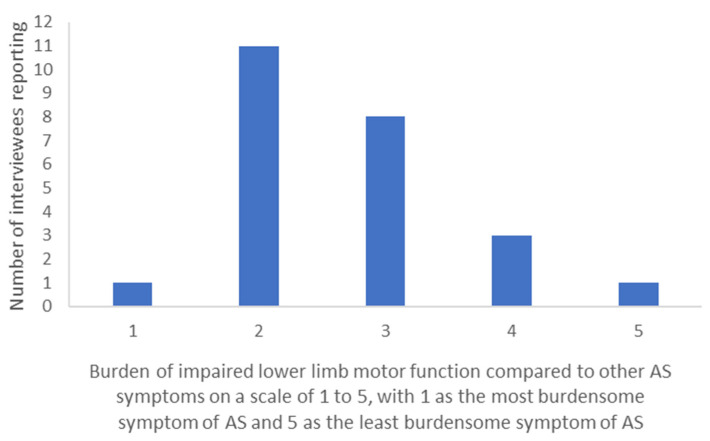
Caregiver ranking (from 1 the most burdensome to 5 the less burdensome) of impaired lower limb motor function burden compared to other AS symptoms.

**Figure 2 children-10-01462-f002:**
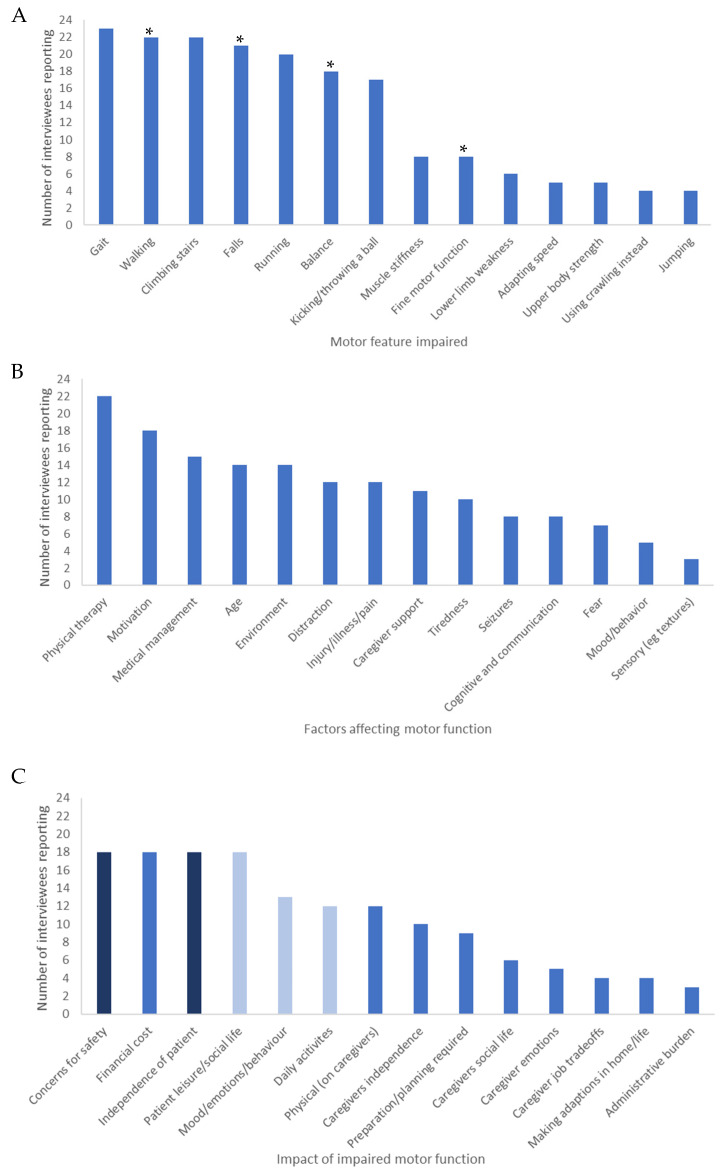
(**A**) Caregiver-reported motor function impairments in AS, (**B**) Factors affecting motor function in AS; (**C**) Impacts of motor impairment in AS. (**A**) Bars with stars (*) indicate concepts that caregivers reported to be the improvement of motor function that would have the greatest impact on their family life. (**C**) Impacts on individuals with AS shown in mid-blue, impacts on caregivers shown in pale blue, impacts on both shown in dark blue.

**Figure 3 children-10-01462-f003:**
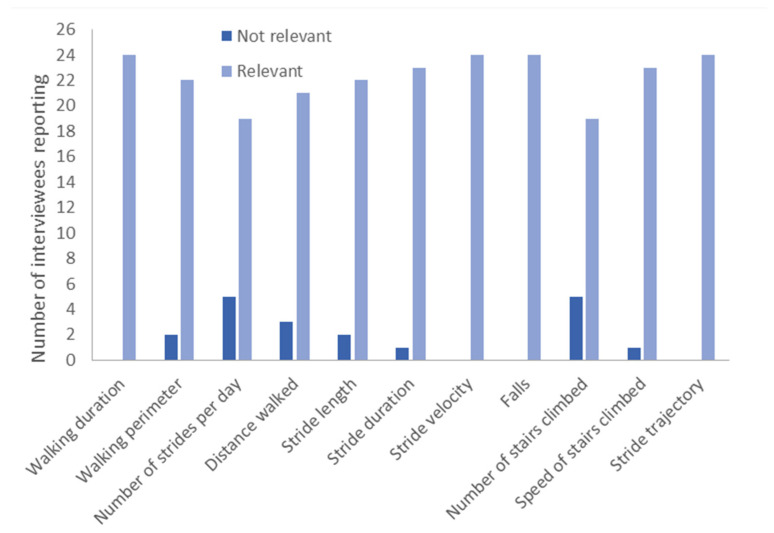
Relevance of potential lower-limb motor function variables to monitor drug effectiveness in AS, as reported by caregivers.

**Figure 4 children-10-01462-f004:**
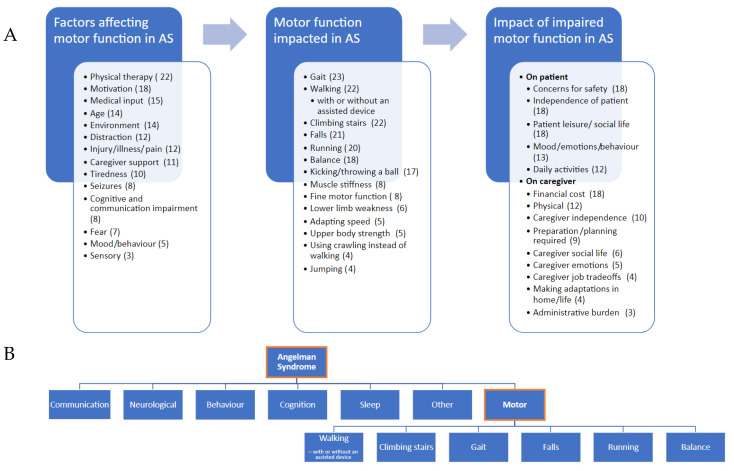
(**A**) A motor-function concept model of AS, identifying impaired motor features in AS, factors affecting motor function and the impact of impaired motor function. Numbers in brackets after each theme represent number of caregivers reporting this theme. (**B**) A conceptual framework of motor function. Categories of AS concepts taken from a previous concept model of AS [8].

**Table 1 children-10-01462-t001:** Sociodemographics of caregivers and individuals with AS.

Characteristic	Caregiver (*n* = 24)	Child/Adult with AS (*n* =24)
Age (years)		
Mean (SD)	41.6 (9.7)	10.8 (8.5)
Median (range)	42.0 (24.0–71.0)	7.5 (2.5–35.0)
Sex, *n* (%)		
Male	5 (21%)	15 (63%)
Female	19 (79%)	9 (37%)
Relationship to individual with AS, *n* (%)		
Mother/Stepmother	16 (67%)	N/A
Father/Stepfather	5 (21%)	N/A
Other *	3 (13%)	N/A
Country of residence, *n* (%)		
USA	12 (50%)	N/A
Chile	3 (13%)	N/A
Belgium	2 (8%)	N/A
Colombia	2 (8%)	N/A
France	2 (8%)	N/A
Argentina	1 (4%)	N/A
Jordan	1 (4%)	N/A
UK	1 (4%)	N/A
Employment status, *n* (%)		
Employed, full- or part-time	18 (75%)	N/A
Homemaker	6 (25%)	N/A
Education status, *n* (%)		
Secondary/high school	1 (4%)	N/A
College degree	18 (75%)	N/A
Postgraduate degree	4 (17%)	N/A
Other **	1 (4%)	N/A
Age at confirmed diagnosis (years)		
Mean (SD)	N/A	2.0 (1.1)
Median (range)	N/A	1.65 (0.6–5)
Ethnicity/race, *n* (%)		
Caucasian	N/A	14 (58%)
Hispanic or Latino	N/A	9 (38%)
Other ***	N/A	1 (4%)
Genotype, *n* (%)		
Deletion	N/A	15 (63%)
Non-deletion	N/A	9 (37%)
Patient is able to speak without technological assistance, *n* (%)		
Yes	N/A	4 (17%)
No	N/A	20 (83%)
Patient is able to walk ****, *n* (%)		
Without assistance	N/A	15 (63%)
With assistance	N/A	5 (21%)
Unable to walk	N/A	4 (17%)
Age when reaching walking milestone (years)		
Mean (SD)	N/A	3.5 (1.5)
Median (range)	N/A	3.0 (1.5–7.0)

* Including a consultant neurologist, an aunt and a sister involved in the care of an individual with AS; ** Master’s Degree; *** Iranian; **** ‘Able to walk’ was defined as ability to ambulate on their feet, with or without assistance.

**Table 2 children-10-01462-t002:** Saturation analysis.

	1st	2nd	3rd	4th
Interviews included	2, 4, 5, 9, 11, 19	20, 21, 22, 23, 13, 14	1, 7, 3, 6, 10, 12	24, 8, 15, 16, 17, 18
New concepts arising in motor features, factors affecting and impact on motor function (in ≥10% of interviewees)	40	2	0	0
		- Adapting speed (motor feature) - Cognitive and communication impairment (factor affecting)		

**Table 3 children-10-01462-t003:** Example quotes to illustrate each impacted motor feature, factor affecting motor function and impact of impaired motor function.

Theme	Impacted Motor Feature/Factor Affecting Motor Function/Impact of Impaired Motor Function	No. of Interviewees Reporting Concept Spontaneously (s) or When Probed (p)	Quote(s) to Illustrate
Impaired motor feature	Gait	16s, 6p	*“She has a very automatic gait. It’s a little as if she were a robot.”* - Mother of a 6-year-old female, France *“He can walk, but he has a significant ataxia gait with broad-based imbalance” * - Clinician discussing an 8-year-old male, USA
	Walking	14s, 8p	“*Well, it impacts her ability to walk efficiently. She doesn’t walk at a normal speed. She walks with a wide gait. She is imbalanced.”* - Aunt of a 20-year-old female, USA
	Climbing stairs	12s, 10p	“*Especially when there are hurdles or climbing stairs, this gives difficulties, and then she needs a hand to assist her.”* - Mother of a 4-year-old female, Belgium
	Falls	13s, 8p	*“Mild to moderate affects balance and he falls more frequently than he might otherwise.”* - Father of a 7-year-old male, Jordan
	Running	7s, 12p	*“She’s not able to run yet, and sometimes when I think she’s trying, she’s walking very fast, and then she looks she’s about to fall over.”* - Mother of a 4-year-old female, USA
	Balance	18s, 0p	*“She has a gait with an enormous lack of balance.”* - Mother of a 6-year-old female, France
	Kicking/throwing a ball	3s, 14p	*“With regard to kicking a ball, he can’t really kick a ball, but if he wants to play with a ball, he needs to be supported by his mother to be able to make a movement that pushes the ball forwards, which isn’t really kicking a ball.”* - Father of a 10-year-old male, U.K.
	Muscle stiffness	8s, 0p	*“He’ll often keep his leg stiff, move it, and then the next leg”* - Mother of a 2-year-old male, France
	Fine motor function	8s, 0p	*“He struggles to pick things up and move them, such as picking up a glass and putting it somewhere else a few meters away.”* - Mother of a 5-year-old male, Belgium
	Lower limb weakness	6s, 0p	*“The entire lower portion of his body is so weak”* - Sister of a 26-year-old male, USA
	Adapting speed	5s, 0p	*“I don’t think that he knows how to control the speed that he walks. It’s the same speed, the same pace, all the time, whereas you and I can walk quite faster, walk faster, walk slower, run, run as fast as Usain Bolt or things like that.”* - Father of a 10-year-old male, U.K.
	Upper body strength	5s, 0p	*“…upper body strength, this girl is stronger than most men I know. I have a friend of mine who used to be one of the UFC champion, and she slapped him so hard that his sunglasses flew off his face 40ft.”* - Father of a 11-year-old female, USA
	Using crawling instead	4s, 0p	*“It’ll be lazy or smart, or she might walk 10ft and then crawl the other 60ft to you.”* - Father of a 11-year-old female, USA
	Jumping	4s, 0p	*“He can walk, but he can’t run or jump or balance on one foot”* - Mother of a 35-year-old male, USA
Factors affecting motor function	Physical Therapy *impact of physical therapy on motor skills*	19s, 3p	*“Through physical therapy, there’s been some progress in terms of, I think, control of movements in terms of building up tone because the issues of hypotonia are very challenging.” * - Clinician discussing an 8-year-old male, USA
	Motivation *impact of motivation on motor skills*	18s, 0p	*“It depends on the motivation. If he has an objective and a motive, he can walk a kilometre. For example, if he sees a truck, he could fall down, grab onto something or continue with his objective. If he doesn’t have motivation, he doesn’t want to walk because he’s not interested.”* - Mother of a 13-year-old male, Chile *“Sometimes I notice when she wants something or there’s motive for her, for instance, I have a cookie in my hand which she loves cookies, there’s more motive for her to go get that cookie. You can see that she’ll walk and show that improvement, but when there’s no motive or not interested, there’s no motivation, not as good with the motor skills.” * - Father of an 11-year-old female, USA
	Medical management *impact of medical management (orthotics, medications, specialised equipment) on motor function*	13s, 2p	*“When he’s wearing his AFOs, that provides more stability for him to walk compared to when he’s not wearing his orthotic devices.”* - Clinician discussing an 8-year-old male, USA *“They take cannabis oil. One of them particularly, when he began with cannabis oil, he used to tremble a lot and wobble. He used to fall over, but then he got better. The cannabis oil made him better.”* - Mother of a 12-year-old male, USA *“Getting control of seizures, on the one hand, helps with safety in terms of walking. On some medications, he’s a little more tired or drowsy after taking the medications, so then maybe he won’t want to walk or will be too drowsy or too low-toned to participate in walking.” * - Clinician discussing an 8-year-old male, USA
	Age *impacts of age on AS motor function*	14s, 0p	*“Yes, and it’s been an improvement just because of her age. She’s getting more skills as she goes.”* - Mother of a 4-year-old female, USA “*As he’s got older, he doesn’t fall over, let’s say, but he’s much more cautious and slower.”* - Mother of a 26-year-old male, Argentina
	Environment *impact of the environment on motor function, including changes in surfaces that are difficult to navigate, to being stuck indoors due to COVID*	14s, 0p	*“What’s hard about that is it depends on the environment. If it was in the home setting and the floor was clear, I would feel better. If there were things in her way, impediments or if we’re outside and there’s uneven surface, I would feel less safe. It depends on the environment.”* - Aunt of a 20-year-old female, USA
	Distraction *impact of being easily distracted on motor function*	12s, 0p	*“Enclosed, where there aren’t lots of distractions and she can be more relaxed and calmer. Her movements are smoother. She’s more attentive to sounds and what is being said to her.”* - Mother of a 4-year-old female, Colombia
	Illness/injury/pain *impact of acute illness, injuries or pain on motor function*	12s, 0p	*“The worst thing is every time she gets sick, we see that there’s a drop and she falls back in her development.”* - Mother of a 4-year-old female, Belgium
	Caregiver support *impact of caregiver support on motor function*	10s, 1p	*“With a helping hand he can probably go a little bit further. On the treadmill, he’s gone up to 10 to 15 min I guess in that pace. When we walked unassisted without any equipment it’s starting and stopping, so is a little bit stand still.”* - Mother of a 6-year-old male, USA
	Tiredness *impact of tiredness/poor sleep on motor function*	10s, 0p	“*He would walk a little bit and then be tired and not want to walk anymore.”* - Clinician discussing an 8-year-old male, USA
	Seizures *Impact of seizures and medication for seizures on motor function*	8s, 0p	*“Her seizure activity might impact her ability to walk. She’s going through a period where she had a lot of seizure activity, she may be more imbalanced.”* - Aunt of a 20-year-old female, USA
	Cognitive and communication impairment *impact of cognitive skills, cognitive development level and communication on motor function*	8s, 0p	*“As he grows, there is also the cognitive side that develops a little more. Therefore, he becomes more curious and has a greater desire to explore and stand up.”* - Mother of a 2.5-year-old male, France *“If we could communicate better and understand each other, we could walk better through life. Otherwise, we are just dealing with the risks or making sure he doesn’t fall.”* - Mother of a 13-year-old male, Chile
	Fear *impact of fear on motor function*	7s, 0p	“*His movements worsen with fear. He’s really afraid of heights. Even to go across this bridge, between here and the conference room on the second floor, he wants a hand to hold because he’s afraid he’s going to fall off the bridge, even though there’s glass walls there. Fear hinders his movement to the point where he will sit down on the ground rather than cross a bridge.”* - Mother of a 35-year-old male, USA
	Mood/behaviour *impact of mood/behaviour on motor function*	5s, 0p	*“I think her mood plays a factor in terms of mobility on occasion, when she gets angry and mad, it seems to be more force in that.”* - Father of an 11-year-old female, USA
	Sensory *impact of sensory features eg changes in texture on motor function*	3s, 0p	*“I feel like he can go longer outside if there’s something he’s looking forward to, walking through the sensory, like the leaves crunching or if it’s snowy. If he feels like there’s puddles to kick around and get wet, then he’ll definitely last longer outside because I think there’s just more distractions and more sensory overload.”* - Father of a 5-year-old male, USA
Impact of impaired motor function	Concerns for safety *impact on safety such as navigating stairs, streets, new environments*	14s, 4p	*“I think it’s a little bit of a double-edged sword because now that he’s able to walk independently. He also has no awareness of safety. He likes to leave or try to run into the street.”* - Mother of a 6-year-old male, USA *“I guess he is more likely to fall like tripping, not even necessarily on obstacles, but like on his own feet, and so it would definitely impact the safety of the fall.”* - Clinician discussing an 8-year-old male, USA
	Financial cost *impact on finances, such as costs of doctors/physios/equipment/home modifications/care*	7s, 11p	*“We’ve definitely had to spend more money on wheelchairs like this particular chair was $800, so it affects that. We have to have a special lift for the van that was $10,000 for the seat to get her in and out. We have to have a van.”* - Mother of a 22-year-old female, USA
	Independence of individual with AS *impact on independence eg to move around by themselves, to go where they want*	11s, 7p	*“She has hardly any independence at all because of her inability to just go where she wants to go.”* - Mother of a 22-year-old female, USA
	AS individual leisure/social life *impacts of leisure/social activities eg going shopping, going to the beach/park etc*	13s, 5p	*“Going places is more difficult, and sometimes things we can do in places that we can go and things that I know she would really like, and because of lack of access for wheelchairs, then there are things that she can’t do. For instance, if there was a swimming pool that we can access or things like that.”* - Mother of a 22-year old female, USA
	AS individual mood/emotions/behaviour *impact on emotions and behaviour, such as anger, upset, frustration*	6s, 7p	*“I think she gets angry that she can’t get somewhere, for example, even in her chair.”* - Father of an 11-year-old female, USA *“He’s a happy child, but sometimes he gets frustrated because sometimes he can’t do things. Like, if all the children are jumping or playing, and he can’t jump, he gets frustrated.”* - Mother of an 8-year-old male, USA
	Daily activities *impact on skills to care for self and undertake everyday activities, such as washing, dressing, eating, brushing teeth, going to school*	8s, 4p	*“In the daily activity, the lower limb function, it’s more on his adls [activities of daily living] getting him in and out of the shower. Again, you’re trying to balance on one foot when the other foot comes up to take a step over that bathtub ledge or the shower ledge, so he needs assistance there.”* - Mother of a 35-year-old male, USA *“She can’t get dressed on [her] own. She can’t brush her teeth on her own, she can’t really clean herself out of her own, personal hygiene, that’s what I mean.”* - Father of an 11-year-old female, USA
	Physical (on caregivers) *physical impact on caregivers, usually relating to needing to pick up/carry individual with AS, resulting in caregiver injuries, pain or concern re getting injured*	11s, 1p	*“My back is gone out right now. Because he’s so heavy and he can’t do things his own, like getting out of the tub. The lifting on us is super cumbersome and it is hard.”* - Father of a 5-year-old male, USA
	Caregiver’s independence *impaired independence of caregiver due to need to be with individual with AS*	8s, 2p	*“She is always with someone, always, always. And it’s the same thing with our independence. For us, as parents to have a little time and independence, we need the help of a nanny.”* - Mother of a 4-year-old female, Colombia
	Preparation/planning required *increased need for planning and preparation eg when going on a walk, going to the beach etc*	9s, 0p	“*We have to consider whether things will be accessible to whatever stroller or thing that we’re using in terms of where we’re going. Will there be room for it? Will there be an elevator? If we’re going to the beach, we have to bring one that has wheels that can go on the beach so that he would be able to participate in those activities.”* - Clinician discussing an 8-year-old male, USA
	Caregivers’ social life *impact on caregivers social life, such as difficulty going out to parties, concerts, family events*	5s, 1p	*“We can’t always go out and do the social things we want because of [name]’s behaviors.”* - Father of an 11-year-old female, USA
	Caregivers’ emotions *emotions resulting from individual with AS impaired motor function, including fear, upset, anxiety, stress, frustration*	5s, 0p	“*The anxiety level goes up for all of us and the intensity with which you have to like walk with him, grab him, catch him. It’s a lot, so yes.”* - Father of a 5-year-old male, USA *“If we can’t take him up the stairs anymore, it makes me very upset to know that I can’t bring him over anymore.”* - Sister of a 26-year-old male, USA
	Caregiver job tradeoffs *job sacrifices due to AS impaired mobility eg giving up job, doing less hours/parttime*	4s, 0p	*“My husband is the only one who works because I have to take care of him.”* - Mother of an 8-year-old male, USA
	Making adaptations in home/life *impact on needing to make adaptations, eg getting special flooring/equipment/car*	4s, 0p	*“I had to remodel our whole entire bathroom and take the bathtub out, because even with grab bars for her trying to lift her leg into the tub, she was so shaky and afraid that she couldn’t make it in there anymore.”* - Father of an 11-year-old female, USA
	Administrative burden *burden resulting from paperwork requirements, contacting insurance companies, sorting claims/medical equipment from said companies*	3s, 0p	*“There’s regular insurance paperwork, which is a minor paper, and then there’s special needs paperwork, which feels like the Mount Everest of paperwork. Constant emails. It’s ridiculous.”* - Father of an 11-year-old female, USA

## Data Availability

The data presented in this study are available on request from the corresponding author. The data are not publicly available due to patient confidentiality.

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
