# Peer review of "Qualitative Insights into Key Angelman Syndrome Motor Related Concepts Reported by Caregivers—A Thematic Analysis of Semi-Structured Interviews"

_children, 2023, doi:10.3390/children10091462_

Round 1

Reviewer 1 Report

Thank you for study.   

The study investigated the motor features of AS in detail from the caregiver's point of view and aimed to understand the impaired motor features and the factors affecting them and the effect of this disorder. I think it will be very useful for future studies.

In the study, it was especially tried to focus on motor influence and the factors affecting it. The authors have created a nice classification for motor involvement. It is possible that walking may be particularly affected by the assistive devices used. For this reason, walking can be specified in two different parts as 'walking with an assistive device' and 'walking without an assistive device' under motor classification subheadings in Fig 4.

Author Response

Thank you for your comments. We agree that walking may be particularly affected by the assistive devices used. We have added this into Figure 4 to more clearly reflect this concept (line 345).

Reviewer 2 Report

The article covers a theoretical and methodological need that would support a digital instrument for the motor control of subjects with Key Angelman Syndrome Motor. Its research objective is: We aimed to develop the motor and 21 movement aspects of the concept models, to support research regarding motor-related digital out-comes aligned with patients’ and caregivers’ perspectives.

The authors are requested to improve the following aspects:

1)      At the discussion section end, a subsection specifying the strengths and limitations of the research is recommended, some of which are scattered throughout the document (Example: Lines: 368-370). It can be emphasized in the small size of the sample surveyed, the subjective analysis of the intervening variables described by the interviewees, the need to empirically quantify and counteract the variables analyzed with tests of greater reliability and precision, with emphasis on new undocumented variables.

2)      In the conclusions section, it is recommended to specify the most important motor variables.

Author Response

Thank you for your comments. We have addressed them as follows:

  1. We have added in a subsection specifying the strengths and limitations of the research (line 414-436)

“Strengths of this study include the thematic analysis, an established method of qualitative data analysis; evidence of concept saturation demonstrating that a suitable number of interviews were used to elicit appropriate concepts; and the input of the patient advocacy group FAST-AS in the development of our interviews and protocol, to ensure that the content of our questions were meaningful to the individuals we aim to support. There were several limitations to this study.  Firstly, our caregiver cohort was selected from those attending the FAST Annual Global Science Summit, meaning that they may not be representative of the overall AS caregiver population. In addition, our caregiver cohort was relatively small and nearly exclusively composed of direct relations of individuals with AS, with one professional (consultant neurologist). Additional work encompassing more professionals involved in the care of AS would be valuable to further develop the concept model. The use of qualitative methods also introduces a degree of subjectivity to the analysis of the variables described by interviewees, so we used two researchers to conduct the coding. Future work should focus on the motor concepts raised in this study that have not been identified in previous work, to further quantify these with greater reliability and precision. Furthermore, as we chose to focus on AS motor features rather than start with a broader exploration of the patient and caregiver experience, there is the possibility that we may have missed some aspects of the context of the motor features within the wider AS symptoms. As mentioned earlier, some motor concepts reported in previous studies did not arise in our interviews, which may have been due to the purely motor-focused approach of our study. To mitigate this issue, we utilized the previously-reported conceptual models to develop our initial analysis plan, and highlighted the similarities and differences between their work and our findings.”

  1. We have added in the most important motor variables to our conclusions section (lines 441-445):

“Furthermore, our work identified 6 important motor concepts (gait, walking, climbing stairs, falls, running and balance) which could serve as potential candidates for novel AS clinical endpoints and our work has provided context to support further investigation and development of these endpoints, using objective measures.”